# Breathable Materials for Triboelectric Effect-Based Wearable Electronics

**Congju Li [1,\*], Ran Cao [1,2,3] and Xiuling Zhang [1,2,3]**

[1]   Beijing Key Laboratory of Resource-Oriented Treatment of Industrial Pollutants,
      School of Energy and Environmental Engineering, University of Science and Technology Beijing,
      Beijing 100083, China; caoranaw@sina.com (R.C.); zhangxiuling@binn.cas.cn (X.Z.)
[2]   Beijing Institute of Nanoenergy and Nanosystems, Chinese Academy of Sciences, Beijing 100083, China
[3]   School of Nanoscience and Technology, University of Chinese Academy of Sciences, Beijing 100049, China
[\*]   Correspondence: congjuli@126.com

**Abstract:** Wearable electronics are believed to be the future of the next-generation electric devices. However, the comfort of current wearable devices is greatly limited due to the use of airtight materials, which may even lead to inflammation of the skin. Therefore, breathable, skin-friendly materials, are highly desired for wearable devices. Here, the recent progress of the breathable materials used to fabricate skin-friendly electronics is reviewed by taking triboelectric effect-based wearable electronics as a typical example. Fibers, yarns, textiles, and nanofiber membranes are the most popular dielectric materials that serve as frictional materials. Metal mesh, silver yarn, and conductive networks made up of nanomaterial are preferred as air-permissive electrodes. The breathable materials for skin-friendly wearable electronics summarized in this review provide valuable references for future fabrication of humanized wearable devices and hold great significance for the practical application of wearable devices.

**Keywords:** wearable electronics; breathable materials; triboelectric effect

## 1. Introduction

The rapid development of electronic technology has brought portable and wearable electronics into a new era [1–3]. Wearable electronics tend to be multifunctional, small, humane, and comfortable [4,5]. Health monitoring has becoming a hot topic in recent years and wearable electronics are one of the most convenient ways to monitor people's health conditions. Wearable electronic with good air permissivity, high sensitivity, light weight, and small volume are highly desirable for real-time and long-term health monitoring. However, commonly used materials for wearable devices are cast films such as polydimethylsiloxane (PDMS) [6–8], fluorinated ethylene propylene (FEP), and metal foils [9,10]. There is no doubt that the assembly of those materials will be stiff and airtight, which obstructs the comfort of human skin or organs. Even worse, long-term wear of such devices may lead to inflammation [11,12]. Therefore, breathable materials such as textiles are promising candidates for the fabrication of wearable devices [13,14].

For the operation of electric devices, a power supply is often necessary. However, currently, power supplies such as bulk and stiff batteries are burdensome for lightweight, convenient wearable electronics. The newly raised triboelectric effect-based devices can generate electric signals under external mechanical force by themselves, which can be regarded as self-powered electronics that are free of battery [15]. With features such as light weight [15,16], simple structure [17,18], easy integration [19,20], and cost-efficient [21,22], triboelectric effect-based wearable electronics have wide applications in the area of energy conversion [23,24], biomedical sensing [25,26], and human–machine

interface [27,28]. However, the comfort of those wearable electronics is still a big challenge. In this review, based on the triboelectric effect, the most recent materials and fabrication of air-permissive wearable electronics are discussed and summarized, which represents a further step toward the fabrication and design of future breathable wearable devices.

## 2. Working Mechanism and Application of Wearable Electronics Based on the Triboelectric Effect

### 2.1. Working Mechanism of the Triboelectric Effect-Based Electronics

There are four operation modes that depict the working process of triboelectric effect-based electronics, namely vertical contact-separation mode, in-plane sliding mode, single-electrode mode and freestanding friction layer mode [29–32]. The electron-generation processes of the four working modes are illustrated in Figure 1. As the principles of the four modes are similar and have been summarized in previous works, only vertical contact-separation mode is explained here in detail as a typical example to get a better understanding of the electron-generation process of the device.

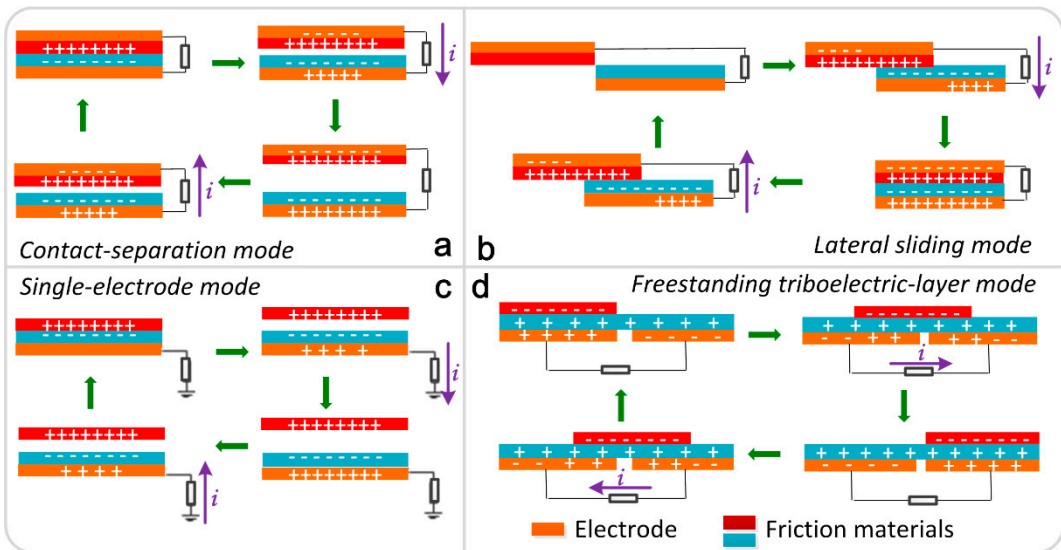

**Figure 1.** Four working modes of triboelectric effect-based electronics. Working mechanism of the device in vertical contact-separation mode (**a**), planer sliding mode (**b**), single-electrode mode (**c**), and freestanding friction layer mode (**d**).

As presented in Figure 1a, in the initial state, the surfaces of the two friction materials will obtain equal amounts of positive and negative charge on the surface of the friction layers when the two friction materials contact each other. This phenomenon can be explained by their different ability to gain or lose electrons, which relies on their varied tribo-polarity [33–35]. Once the two layers move away from each other, the electron will move through the external circuit to balance the potential difference between two electrodes attached on the back of friction layers (between the electrode and the ground in single-electrode mode). Therefore, the current and voltage can be obtained in the eternal circuit and further be used. When the two friction layers are approaching, the current in the opposite direction will appear in the external circuit until the two layers full contact each other. Then, the next working period begins and repeats the whole process.

### 2.2. Application of Triboelectric Effect-Based Wearable Electronics

As mentioned above, triboelectric effect-based wearable electronics have wide applications. With high sensitivity, the electronics can be used as self-powered sensors to monitor human health ranging from elbow motion and finger bending to breath and heartbeat [36,37]. Moreover, triboelectric effect-based wearable devices can perform human–machine interaction, which could

easily control computer and home applications [38,39]. Figure 2 presents the overview application of wearable devices in energy conversion [40,41], health monitoring [42,43], pressure detection [44], and human–machine interaction [45]. From the point of comfort and humane use, the use of breathable materials is a promising direction to optimize the development of wearable devices, as materials are the fundament of the device, which will be discussed in the following section.

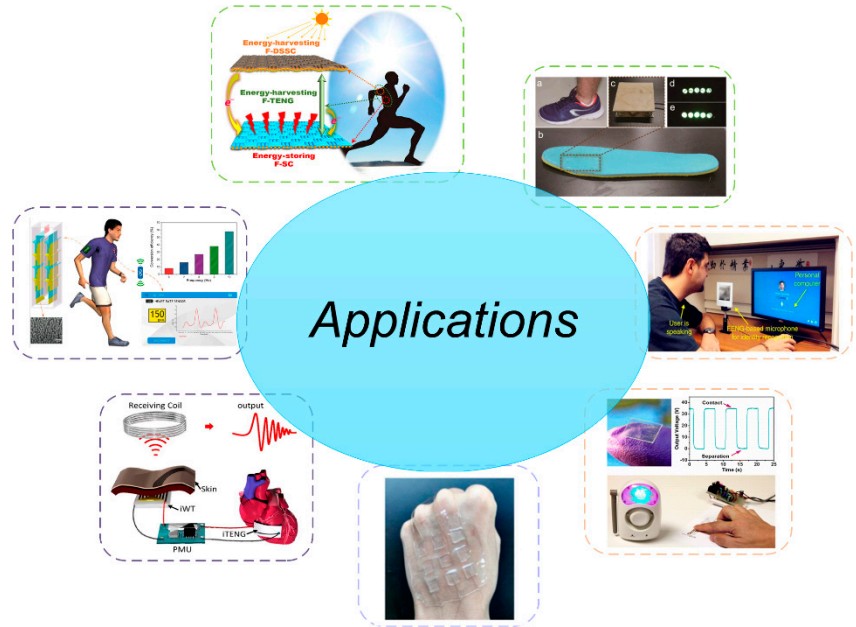

**Figure 2.** Applications of triboelectric effect-based wearable device, including energy harvesting, health monitoring and human–machine interfacing. Figure adapted from ref. [39–45].

## 3. Breathable Wearable Electronics Based on Triboelectric Effect

### 3.1. Woven-Structured Wearable Electronics

Everything around us has friction characteristics. As textiles and fibers are believed to be the second human skin [46,47], researchers have considered using fibers, the essential components of clothes, to fabricate triboelectric effect-based wearable devices. Zhou was the first to put this idea into action [48]. By weaving polyester and nylon fabric, the triboelectric effect-based wearable device is breathable, flexible, and washable, which is similar to commonly worn clothes. Figure 3 is the structure illustration and fabrication process of the woven-structured electronics. To realize the washability and flexibility, silver fabric was chosen to replace metal foils to act as the electrode of the device. A single component of the device was constituted through covering both sides of the silver fabric with nylon or polyester. Weaving the single component of the fabrics together finished the device. In this situation, nylon and polyester fabrics act as two friction materials of the device. The working mechanism of the device can be concluded to freestanding friction layer. Benefiting from the easy integration of the device into clothes such as shirts, trousers or shoes, the device can harvest different kinds of human motion energy and powers several LEDs effortlessly. This work opened a new approach for fabricating breathable and comfortable wearable electronics.

Inspired by the components of clothes, the homemade woven-structured device is a great step forward in skin-friendly wearable devices. However, the fabrics were weaved by hand, which is time-consuming and relatively expensive. Therefore, a mass-production method is necessary to realize the commercialization of the wearable device [49]. To solve this problem, normal yarn-winding machine was introduced to fabricate wearable electronics.

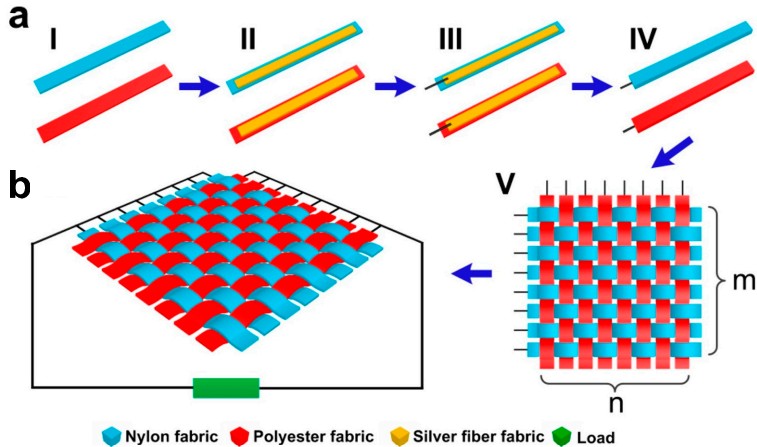

**Figure 3.** Woven-structure wearable triboelectric devices. (**a**) Fabrication process of the wave structure device. (**b**) Structure and circuit of the wave structure device. Figure adapted from ref. [48].

Recently, a wearable triboelectric device (WTD) made of compound yarns, mass produced by a commercial yarn-winding machine, was put forward [50]. As presented in Figure 4a, the yarn has a core-shell structure, where silver-plated yarn and nylon-6 (PA6) or polytetrafluoroethylene (PTFE) acts as core and shell, respectively. During the contact-separation process, PA6 and PTFE will rub against each other and silver-plated yarn serves as an electrode to collect the induced electrons. Figure 4b is a simple illustration of the process of how the friction materials such as PA6 or PTFE covered the surface of the silver-plated yarn through the winding machine. The flexibility and washability of the WTD are shown in Figure 4c, which shows firm evidence for the practicability of the WTD when integrated with commonly worn clothes. Moreover, the cross-sectional and top view of the core-shell yarn was presented in the bottom of Figure 4c as well. When directly integrated into the garment, the WTD with a size of 4 × 4 cm can easily charge a capacitor from 0 to 5 V within 100 s and, further, power an electric watch and scientific calculator. With the help of the commercial yarn-winding machine, a breathable, flexible, and washable WTD can easily be scaled up, which is a significant step forward for the mass fabrication and commercialization of wearable WTDs.

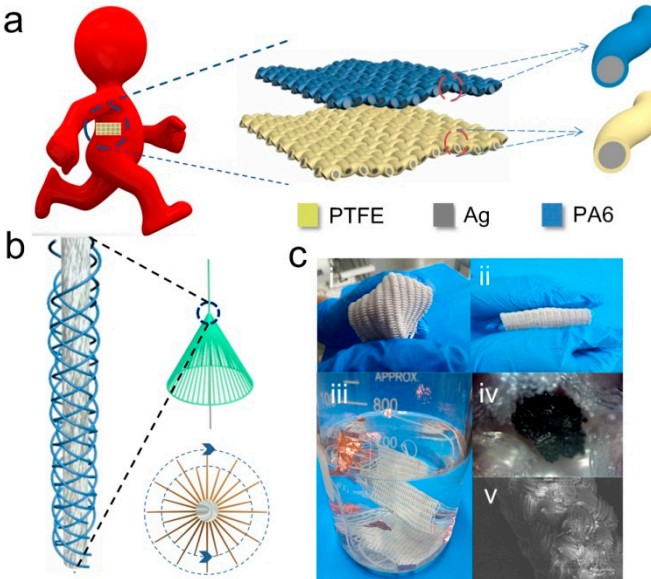

**Figure 4.** Wearable device made up of compound yarns, mass produced by a commercial yarn-winding machine. (**a**) Structure and application of the wearable triboelectric device (WTD). (**b**) The schematic of the formation of one yarn. (**c**) The flexibility and washability of the WTD and cross-sectional view of one yarn (Scale bar, 2 cm (i–iii), 200 μm (iv), 400 μm (v)). Figure adapted from ref. [50].

### 3.2. Textile-Based Wearable Electronics

The fabrication process of the fiber-based wearable electronics is still complex, as every fiber should be treated seriously. For the purpose of simplifying the fabrication process of the wearable device, Cao et al. directly used textiles to manufacture wearable electronics [38]. A washable electronic textile (WET) can be regarded as a self-powered sensor for human–machine interaction. Figure 5a is a schematic of the structure of the WET. Nylon and silk act as substrate and friction layers, respectively. To guarantee the air permissivity of the whole device, a conductive network made up of carbon nanotubes (CNTs) sandwiched between silk and nylon textiles serves as an electrode. Please note that the conductive network was fabricated through screen-printing CNT ink onto the nylon substrate, which is an easy mass-production method. Moreover, the size and shape of the electrode is a designable benefit of a screen-printing technology. Figure 5b is the optical photograph of the as-fabricated WET with strip array electrode. The scanning electron microscope (SEM) image of the nylon textile covered with CNTs is illustrated in Figure 5c. The magnified image of the surface of the fabric with CNT showed that even grooves between fibers were covered with CNT (Figure 5c), which provides further evidence for the good conductivity of the printed electrode. It is believed that nylon and silk textiles are breathable for human skin. The addition of the CNT conductive network will decrease the air permissivity of the nylon textiles, but the influence is negligible. For example, the air permissivity of the nylon substrate with 20 μm CNT ink on its surface is 88.6 mm/s, which is much higher than jeans (26.4 mm/s). When anchored on the surface of a wrist band, the WET can control software on computers and wirelessly trigger home appliances. Based on commonly worn clothes and screen-printing technology, this work provides a new and easy method to fabricate breathable, washable, and wearable devices for practical applications.

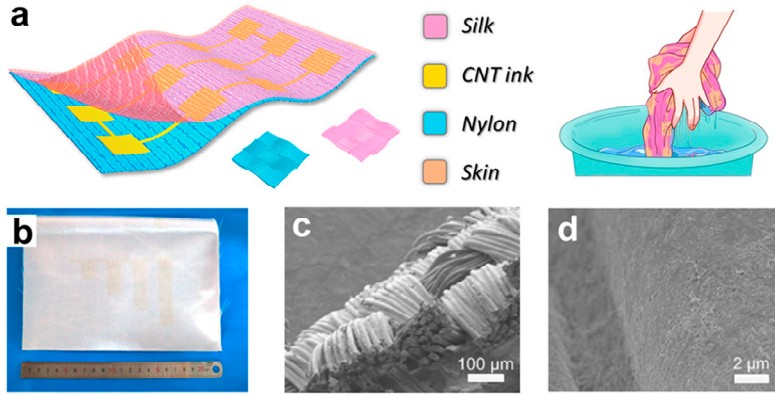

**Figure 5.** The textile-based wearable electronic device. (**a**) Illustration of the structure of the washable electronic textile (WET). (**b**) Photograph of the as-fabricated WET. (**c**) The SEM image of the nylon textile covered with CNT ink. (**d**) The magnified SEM image of the nylon textile covered with CNT ink. Figure adapted from ref. [38].

### 3.3. Nanofiber Membrane-Based Wearable Electronics

The direct use of textile is a convenient and easy way to design comfortable and wearable electronics. Nevertheless, the electric output performance and sensitivity of these electronics are barely satisfactory. According to existing literature, the nanostructure on the surface of the friction materials will increase the output of the triboelectric effect-based devices, as a larger effective contact area will be obtained [51,52]. Therefore, a nanofiber membrane, which has both air permissivity and a nanostructure on its surface, was introduced to prepare the wearable electronics.

As presented in Figure 6 [53], polyvinylidene fluoride (PVDF) and thermoplastic polyurethanes (TPU) nanofibers, prepared through electrospinning, were chosen as friction layers of the device. Figure 6b,c present the SEM images of the surface of the PVDF and TPU nanofiber. The nanofiber membrane was formed by the pile of millions of nanofibers, which lead to nano-micro-holes between

nanofibers. It is speculated that gas can diffuse through the holes. The surface is relatively rough, as well, because of the irregular pile of the nanofibers. At the same time, conductive gauze, and silver elastic textile, which act as breathable electrodes, were used as substrates to collect PVDF and TPU nanofibers. Consequently, the whole device is breathable and comfortable. Interestingly, the TPU membrane and its substrate, silver textile, are elastomer, which made the device stretchable. The stretchability and flexibility of the device is favorable for harvesting irregular human motion such as stretching or twisting the wrist, and is a more comfortable and durable method compared to traditional non-elastic materials.

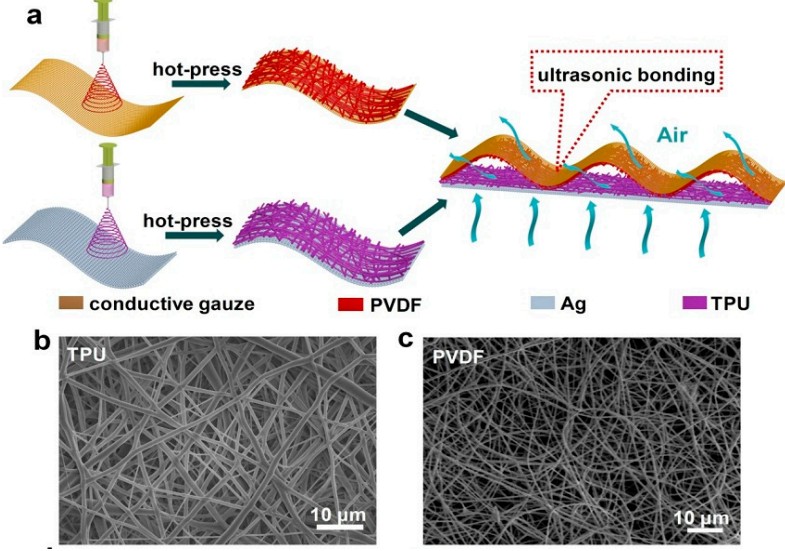

**Figure 6.** Nanofiber membrane-based wearable devices. (**a**) Illustration of the fabrication and structure of the device. The SEM image of the TPU (**b**) and PVDF (**c**). Figure adapted from ref. [53].

As mentioned above, nanofiber membrane-based electronics have high sensitivity compared with textile-based sensors. It has been demonstrated that sensors fabricated with nanofiber membranes were able to detect slight mechanical force. For example, our group put forward a self-powered and nanofiber-based triboelectric sensor (SNTS) for respiratory monitoring [54]. Figure 7a presents the structure of the SNTS. The friction material and electrode is PVDF nanofiber membrane (PVDF NM) and Ag nanoparticles (AgNPs), respectively. The SEM in Figure 7b is the surface of the PVDF NM coated with AgNPs, where little holes indicate the air permissivity of the device in terms of the micro-nano morphology of the materials. Based on a single-electrode working mode, the detailed process of the transfer of the electrons is described in Figure 7c. With the idea of simple structure and mass production, electrospinning and screen-printing are combined to fabricate the SNTS (Figure 7d). The SNTS used for respiratory monitoring, with light weight and small volume, is shown in Figure 7e. Figure 7f compares the air permissivity of the as-fabricated sensor with other commonly used materials. Although the gas permeability of the SNTS is worse than jeans, it is about 4.5 times higher than A4 printing paper, not to mention its superiority compared with cast film. Given features such small volume and light weight, the SNTS can be easily attached on the inner side of a mask to monitor respiratory conditions with high sensitivity.

In previous work, nanofibers of PVDF NW was shown to be waterproof with high electronegativity. However, its application in wearable electronics is still unsatisfactory, as it will crack under high stretch stress or deformation [55]. Therefore, devices which can be stretched, bent, and compressed are necessary to meet the higher requirements of wearable electronics [56]. In addition, for an elastic electrode, conductive networks made up of nanotubes are much better than those made up of nanoparticles, because of the weak interaction and connection between nanoparticles.

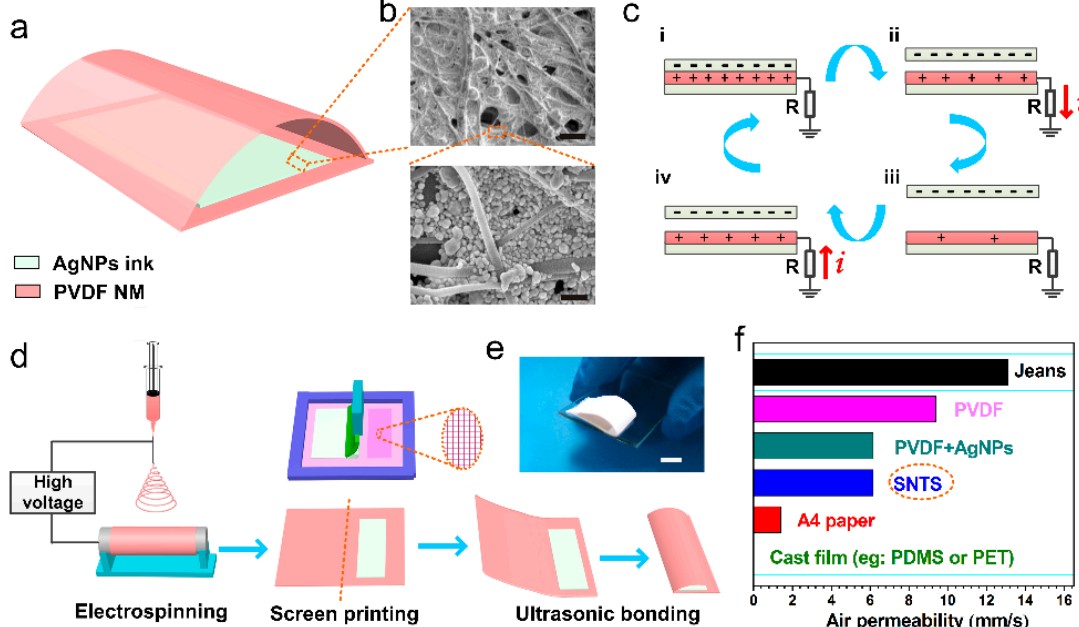

**Figure 7.** Nanofiber membrane-based highly sensitive wearable sensor. (**a**) Structure of the nanofiber-based triboelectric sensor (SNTS). (**b**) The SEM image of the surface of the PVDF NW (scale bar, 3.75 μm and 300 nm). (**c**) Working mechanism of the SNTS. (**d**) Fabrication process of the SNTS. (**e**) Photograph of the as prepared sensor. (**f**) The air-permissive of the sensor and some other materials. Figure adapted from ref. [54].

The structure of a stretchable wearable device (SWD) is shown in Figure 8a [57]. Elastic TPU nanofiber membrane (TPU NM), which has great stretchability and flexibility, was the main component of the device. On the one hand, TPU NM acts as the substrate to support the screen-printing CNT ink. On the other hand, TPU NM was the substrate that holds PVDF NM, the friction layer of the SWD. PVDF NM is used to improve the electric output performance of the SWD, as PVDF is more tribo-negative than TPU. Interestingly, during the electrospinning process, it was found that the nanofiber-collecting substrate is a critical factor that influences the tensile property of the TPU NM. As shown in Figure 8b, the stretchable ability of the TPU NM collected on the substrate of PDMS is much better than that of polyethyleneterephthalate (PET) or polypropylene (PP). The high tensile performance of the TPU NM collected on the PDMS substrate can be attributed to the homogeneous and smooth surface of the PDMS, which avoids the non-uniformity of the NM. Figure 8c is the fabrication process of the SWD, which constitutes the commercial technologies of electrospinning and screen-printing. The breathable and stretchable SWD showed outstanding performance in joints of body. For instance, the as-fabricated SWD can be used to prepare a glove and detect the motion state of five fingers at the same time. The lightweight, air-permissive, and stretchable SWD is more shape-adaptive and skin-attachable, which is of great importance to the use of wearable electronics with high sensitivity.

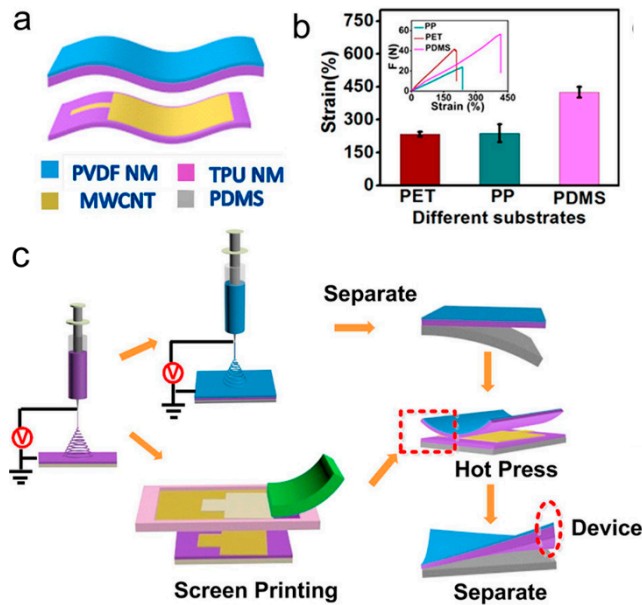

**Figure 8.** Nanofiber membrane-based stretchable wearable sensor. (**a**) Schematic illustration of the structure of the stretchable wearable device (SWD). (**b**) The tensile property of the TPU NM collected on different substrates. (**c**) Manufacture process of the SWD. Figure adapted from ref. [57].

## 4. Discussion

The breathable materials summarized in this review comprise three types, namely fiber, textile, and nanofiber membrane-based wearables. Table 1 presents an overview comparison of the three types of materials. For breathability, fiber and textile-based devices have superiority over nanofiber membrane-based electronics. In contrast to textile and nanofiber membrane, the fabrication process of the fiber-based device is much more complex, as every single fiber is made up of an electrode and friction layer. For electronic devices that have higher requirements with sensitivity, the nanofiber membrane cannot be replaced by fiber or textile-based devices, which benefit from its nanostructure on its surface. Moreover, a nanofiber membrane is soft, lightweight, small, shape-adaptive, and easy to integrate. Therefore, for practical application, sensitivity, breathability, processability, and cost should all be taken into consideration when designing and fabricating wearable electronics.

**Table 1.** Comparison of the breathable materials.

|  | Breathability | Processability | Materials Cost | Sensitivity |
|---|---|---|---|---|
| Fiber-based | 🙂 | 😐 | 🙂 | 🙁 |
| Textile-based | 🙂 | 🙂 | 🙂 | 😣 |
| Nanofiber membrane-based | 😐 | 😐 | 🙂 | 🙂 |

To sum up, Figure 9 depicts the summary of the breathable materials used to fabricate comfortable wearable electronics by replacing traditional airtight materials. To meet the requirements of comfort and skin friendliness, the components of clothes such as fibers and textiles were selected to fabricate wearable electronics, which are also low-cost and easy to integrate with clothes. Wearable electronics fabricated with nanofiber membranes are preferred due to relatively good air-permissibility and micro-nano structure, which guarantees both skin friendliness and sensitivity of the device. Single conductive fibers, metal mesh, conductive textiles, and networks made up of conductive nanomaterial

are the commonly used air-permissive electrodes. Elasticity is another critical factor that influences the comfort of the devices, being is shape-adaptive and thus offering a better attachment to skin.

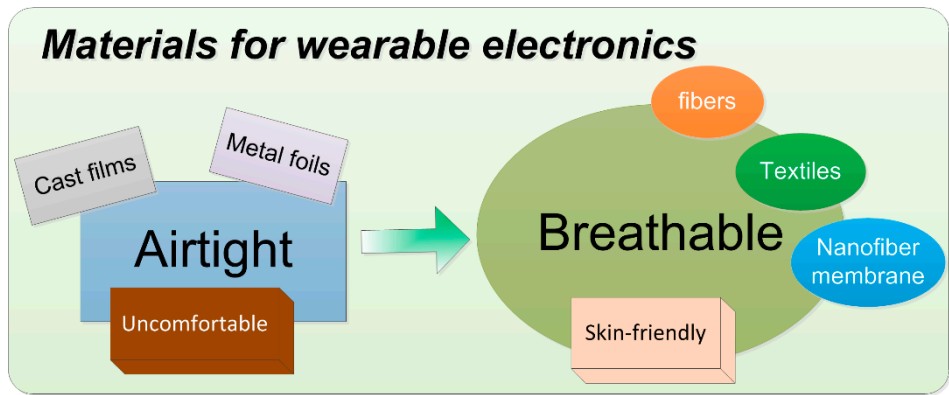

**Figure 9.** Summary of the materials required for triboelectric effect-based wearable electronics to change from airtight layers to breathable materials.

The breathable materials and fabrication methods reviewed in this paper can be expanded to all wearable devices, which provide valuable references for the design and fabrication of skin-friendly devices. However, effects are still needed to realize superior air-permissibility, higher sensitivity, and better stretchability of wearable electronics. Mass production, minimization and easy integration are also critical for the commercialization of wearable devices.

**Author Contributions:** C.L. has proposed the structure of the review and wrote the manuscript; R.C. and X.Z. finished the manuscript together.

**Funding:** This research was funded by the Programs for Beijing Science and Technology Leading Talent (Grant no. Z161100004916168), and the Fundamental Research Funds for the Central Universities (No. 06500100), and the "Ten thousand plan"—National High-level personnel of special support program, China.

**Conflicts of Interest:** The authors declare no conflict of interest.

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
