# Peer review of "Breathable Materials for Triboelectric Effect-Based Wearable Electronics"

_applsci, doi:10.3390/app8122485_

Reviewer 1 Report

The review would like to describe breathable materials used for triboelectric effect in wearable electronics.

However, the structure of the review appears confused.

It is not specify what stand for breathable materials. Seems that each fabric is a breathable materials, so why authors talk about breathable materials instead on simple focus on fully textile sensors? Indeed, just two cited papers present a real value for breathability of the new materials, comparing it with jeans, that it is proposed as no breathable material. In details, page 5 line 144 it is mentioned that CNT on nylon allow an air-permissive of 88.6 mm/s while jeans is just 26,4 mm/s. From this reference [38] seems that jeans is considered as  a low air-permissive fabric. On the opposite, on page 7 line 190, the material SNTS demonstrates a breathability 4.5 times lower than jeans.  Thus it can be considered breathable or not? Moreover, no others examples cited, report any value about breathability. The authors state that the holes presents between the nanofibers allow a good air-premissivity. But it is not demonstrated. I think that, if the authors want to talk about breathability they have to identify reference values to clarify the concept, explain how this value can be measured and report papers in which this aspect has been evaluated.

The working mechanism described in page 2 it is, in my opinion, not described in a proper scientific way. Also in the text there are sentences that are not scientific. Page 5 line 141, pag 8 line 200. 

Why the authors talk about skin and organ? I don’t think that textile wearable sensors would be used as implantable sensors.

In the abstract the authors mentioned the inflammation of the skin problem of airtight materials, that can be overcome with breathable materials. I don’t see the relationship between breathable and no skin inflammation. Indeed it has been demonstrated that CNT ink is toxic, or Ag nanoparticles mentioned in page 7 are toxic. We have less information about nanofibers, but in general nano materials and nanocompounds can penetrate into the skin. The biocompatibility of this new composite has yet to be evaluated.

Moreover the wearable sensors based on triboelectric effect, in theory do not cover all the body, thus I don’t see the great limitation of airtight materials. 

I think that the paragraph on page 6 from line 159 to 170 has to be moved on the next section “Nanofiber membrane”. Then, it is not clear the introduction of this paragraph, respect the previous ones.

In conclusion I think that the present review do not give a clear overview of new breathable materials for wearable sensors based on triboelectric effect. 

Author Response

Thank you for your professional suggestion. As researchers more likely focus on wearable electronics with properties like multifunction, high sensitivity, easy fabrication, miniaturization and so on, breathability has been seldom considered. As far as I am concern, breathability of wearable electronics is a newly raised research direction. Therefore, there are no systemic research results about the breakability of wearable electronics and no related review as well. 

  Indeed, we use several examples of textile to regard as breathable materials. However, we also discussed some electronics fabricated with nanofiber membrane, which may not be classified into textiles, so we used breathable materials instead of textile wearable electronics as the title of this review. When talking about the breathability of the materials, it is true that the air-permissive of the textile is much higher that nanofiber membrane. In page 5 lines 144, we tried to prove that in regard to air-permissive, the influence of the CNT network’s on the textile of the air-permissibility is acceptable. We just regarded jeans as an reference and we did not mean it was a low air-permissive fabric On page 7 line 190 the material SNTS demonstrates a breathability 4.5 times higher than A4 printing paper, but worse than jeans. The main reason that nanofiber membrane chose to fabricate wearable electronics is that it has high sensitivity, small volume, lightweight, softness, and easy to realize miniaturization. To some applications, in contrast with breathability, sensitivity plays a more important role. Therefore, nanofiber with air-permissibility is introduced in assembling wearable electronics. Moreover, even though its air-permissibility is limited compared with clothes we wear, it is much better than airtight materials to some extent, which is more skin-friendly. 

  As mentioned above, breathability of electronics has no clear standard because researchers pay less attention in this aspect. We tried to find more about the value and reported about breathable wearable electronics but failed. We here summary some of the recent progress and hope that more people would take this factor into consideration when designing and fabricating wearable electronics.

The air-permissibility of the nanofiber memebrane was measured by a TEXTEST AG (FX 3300) at a test area and pressure of 5 cm2 and 100 Pa. The numerical of the test machine proved the air-permissive of the nanofiber membrane and the SNTS. As a consequence, we speculate that the holes on the surface of the membrane contribute to the air-permissibility of the membrane, as there are spaces between nanoparticles. We have added “speculate” in the revised manuscript.   

  The working mechanisms of the triboelectric effect based electronics have been described a lot in previous work, especially in reviews. Therefore, here we spent more paragraph in discussing breathable materials rather than explain its working principles. The expression in this paragraph has been revised. The sentences in Page 5 line 141, Page 8 line 200 have been rewritten as well (see revised manuscript). 

  We talked about skin and organ to demonstrate the flexible and shape-adaptive of wearable electronics is necessary for practical application. Even through nanofiber membrane has potential application in implantable sensors, there is no report about the textile sensors used as implantable sensors. Thank you for pointing our neglect and we have deleted “organ” in the revised manuscript. We used the relationship between breathable and no skin inflammation by reference the article “Inflammation-free, gas-permeable, lightweight, stretchable on-skin electronics with nanomeshes” published on Nature Nanotechnology (ref. [12]). May be there is no direct relationship between them, but breathable materials seem better than airtight materials for long time wearing. To be honest, we are not clear whether CNT and Ag nanoparticles are toxic or not. Therefore, in our previous work, electrode assembled with CNT or Ag nanoparticles are sealed inside of the sensor, which will not directly contact with skin. Electronspinning nanofiber membranes have been widely used in medical research, wearable electronics and so forth. To the best of our knowledge, we haven’t seen reports about its toxic. 

  All the examples in this review are were working based on triboelectric effect. We admit that airtight materials are able to satisfy the basic need of current wearable electronics. However, we aim at improving the overall performance of the sensors by take comfortable and skin-friendly materials into consideration.

  The paragraph on page 6 from line 159 to 170 mentioned the silver textile so we put this paragraph following the textile part. Indeed, this content is more related with nanofiber membrane. We have moved this paragraph to the section of “Nanofiber membrane” and thank you very much for your helpful suggestions. Besides, the introduction of this paragraph has been rewritten as well.

Reviewer 2 Report

I have only a small suggestion about the paper

Some typos has been found, such as: line 89 sliver=silver?

Image quality should be improved.

In mi opinion lines 176 to 179 should be in the introduction .

Reference numbers should be before point. Replace .[xx] by [xx].

Author Response

1. Thank you very much for your kind reminder, we have corrected the corresponding typos. 2. We have replaced the previous images with high quality images. 3. We have moved the context from lines 176 to lines 179 to the Introduction section and rewrote this paragraph. 4. Thank you for your careful revision, we have put all the reference number before point.

Reviewer 3 Report

The review lists the current technology of textile triboelectrics. The scope of the technology is well set and includes various approaches to realize textile based triboelectrics. The manuscript needs to be improved for clear categorization and meaningful comparisons of each.

1. Titles of figures should be properly described, relevant to each figure and focused. Figures can be combined in the same category or the figure titles need to describe figure contents more precisely. 

2. It is better to focus on more breathable designs and materials than triboelectrics, according to the title of this review.

3. Figure 3b is missed.

4. Is it possible to provide any comparison of each method to fabricate breathable triboelectrics in terms of breathability, processability, materials, power generation and so on.

5. Figure quality needs to be improved to higher resolution.

Author Response

Thank you for your professional advice. The structure of the review has been reorganized and we have added one paragraph to compare the difference of these breathable materials.

1.      We have rewritten some of the titles of the figures to describe figure contents more precisely.

2.      Thanks for your suggestions. We were invited to write an article that related with triboelectronics. Therefore, we used some paragraph to discuss the triboelectric effect based electronics and explain its mechanism and applications.

3.      We have edited Figure 3 and thank you for reminding us.

4.      Thank you for your very useful suggestion. We have added a table and a paragraph in the revised manuscript to compare these methods and a paragraph to describe the difference between them.

5.      Higher resolution images have been replaced the previous ones.

Round  2

Reviewer 1 Report

Dear authors, 

I appreciate your response and modification on the text. 

I think that now the review is more clear and the cited work are more homogeneous combined.